# Ethological principles predict the neuropeptides co-opted to influence parenting

Christopher B. Cunningham[1], Majors J. Badgett[2,3], Richard B. Meagher[1], Ron Orlando[3,4] & Allen J. Moore[1]

Ethologists predicted that parental care evolves by modifying behavioural precursors in the asocial ancestor. As a corollary, we predict that the evolved mechanistic changes reside in genetic pathways underlying these traits. Here we test our hypothesis in female burying beetles, *Nicrophorus vespilloides*, an insect where caring adults regurgitate food to begging, dependent offspring. We quantify neuropeptide abundance in brains collected from three behavioural states: solitary virgins, individuals actively parenting or post-parenting solitary adults and quantify 133 peptides belonging to 18 neuropeptides. Eight neuropeptides differ in abundance in one or more states, with increased abundance during parenting in seven. None of these eight neuropeptides have been associated with parental care previously, but all have roles in predicted behavioural precursors for parenting. Our study supports the hypothesis that predictable traits and pathways are targets of selection during the evolution of parenting and suggests additional candidate neuropeptides to study in the context of parenting.

[1] Department of Genetics, University of Georgia, Athens, Georgia 30602, USA. [2] Department of Chemistry, University of Georgia, Athens, Georgia 30602, USA. [3] Complex Carbohydrate Research Center, University of Georgia, Athens, Georgia 30602, USA. [4] Department of Biochemistry and Molecular Biology, University of Georgia, Athens, Georgia 30602, USA. Correspondence and requests for materials should be addressed to A.J.M. (email: ajmoore@uga.edu).

The selective pressures that lead to the evolution of parental care are well documented[1]. Parental care typically evolves to minimize unusually stressful or hazardous environments for offspring[2–4]. Although this hypothesis is widely supported[4], parental care is not the only evolutionary solution to adverse conditions. Moreover, it may not be the most likely response as the evolution of parenting reflects changes in multiple behavioural inputs, involving many pathways[5]. At a minimum, the evolutionary transition from asociality (social interactions limited to mating) to subsociality (extensive social interactions between parents and offspring involving parental care) is predicted to require modification of several contributing behaviours including tendencies for dispersal, feeding, mating, aggression and tolerance of social interactions[1–3]. Caring parents no longer disperse from a mating site, they provision food rather than feed themselves, they pause reproduction and mating, they show aggression to protect offspring and shared resources rather than their own resources, and they tolerate the presence of others and increased social interactions[1–4,6]. The early ethological literature therefore predicts that parental care evolves only when there are suitable behavioural and ecological precursors present within the evolutionary ancestor, such as nest building, defensive postures and aggression, and potentially shared resources[2,3].

The early predictions of the specific constituent behaviours were made without reference to the mechanistic changes that would be required. Implied, however, is that repurposing existing traits involves changes in the timing and direction of interactions. This suggests a potential mechanism: that changes in timing of gene expression are involved in the evolution of derived behaviour[1]. If true, we can use the predicted behaviours to be modified from non-parenting to parenting to infer the underlying mechanisms. We specifically hypothesize that modifying behaviours affecting parenting will result from altered gene expression rather than the evolution of novel genes. This hypothesis is a natural extension of Wright's theory of nearly universal pleiotropy[7], suggesting that genes gain functions when used in novel contexts, and the ubiquity of changes of gene regulation that are seen during evolutionary transisitions[8–10]. Moreover, this is consistent with previous work that uses the nature of selection to predict the genetic changes underlying the evolution of social behaviour[1,6,11,12]. For example, when parenting, animals are typically selected to be unreceptive to mating. We therefore predict that mechanisms that influence mating will be altered. Following this logic, overall we predict that parenting will involve changes in expression of genes that influence mating, feeding, aggression, and increased tolerance for social interactions as these are the behaviours modified as lineages evolve from asocial to subsocial[1–3]. However, following Wright[7], we also predict polygenic changes rather than one or few genes. Thus, we need to use techniques that can detect multiple changes.

Studies of changes in gene expression have been revolutionized by an ability to assay overall transcriptional changes in many genes simultaneously. However, transcriptomics is not a particularly powerful method for identifying changes in expression of many genes that code for proteins that influence behaviour, such as neuropeptides, that have low gene expression[13], highly restricted sites of release[14], and can be hard to detect with transcriptomic studies that are not highly tissue specific[15]. Proteomics provides a complementary approach that overcomes some of these limitations and provides a method to target protein categories of interest. Adopting a complementary approach is necessary because neuropeptides strongly influence the social behaviour of animals[16] and many neuropeptides are likely to be associated with parenting. One of the most studied neuropeptides, oxytocin, is necessary for parenting across the animal kingdom[17]. There is a causal relationship between the neuropeptide *galinin* and parental care in mice[18]. In the burying beetle *Nicrophorus vespilloides*, *neuropeptide F receptor* is differentially expressed between parenting and non-parenting states[19]. The importance of neuropeptides is expected, as parenting individuals must undergo many rapid shifts in behaviour. Neuropeptides can exhibit their influence within minutes, have highly localized effects targeting very select neural circuits, or have highly widespread effects targeting many and diffuse neural circuits[14].

Here we test the hypothesis that a transition from a non-parenting state to a parenting state will reflect differences in neuropeptides known to be generally associated with mating, feeding, aggression and increased social tolerance and that neuropeptides influencing other traits will not change during parenting. To test this, we estimated the abundances of neuropeptides of the burying beetles *N. vespilloides* sampled from a solitary and parenting states. *N. vespilloides* adults are normally solitary but switch to parenting in the presence of appropriate resources. Parenting in this species is extensive and elaborate. Adult beetles are stimulated to parent after they locate a vertebrate carcass and bury it. Parents remain on this carcass and provide indirect care by removing the fur or feathers and forming a nest within the carcass. They also repeatedly coat the carcass with excretions that retard microbial growth. Eggs are laid in the surrounding soil, hatch and larvae crawl to the crypt where they interact with one or both parents. Direct parental care involves feeding larvae pre-digested carrion by regurgitation for the first two days of larval life (Fig. 1). Parenting occurs for 75% of larval development, yet lasts only days[20]. *N. vespilloides* is also molecularly tractable with a published genome[21], allowing for efficient proteomic work.

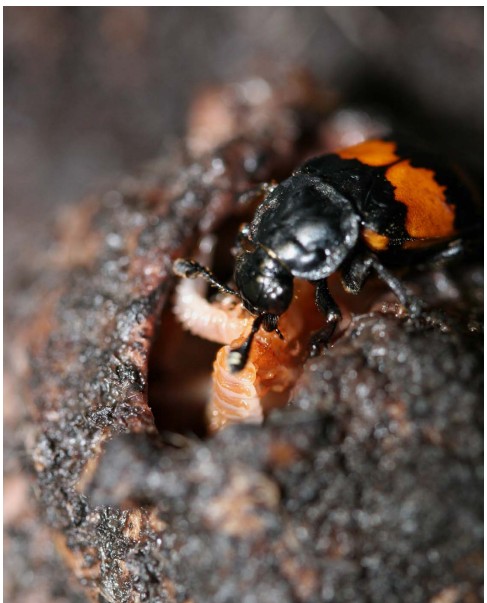

**Figure 1 | A female burying beetle feeding her offspring.** In this species, a parent spends around 72 h preparing a carcass, after which larvae hatch and arrive at the carcass. Once larvae arrive, parents spend a further 72 h feeding larvae (with peak parenting 12–24 h after larval arrival), and then disperse around 100 h after larvae first arrive on the carcass. Larvae disperse fully grown around 125 h after their arrival on the carcass. As shown here, feeding involves direct mouth-to-mouth contact and a transfer of pre-digested carrion from the parent to the offspring. Photograph by AJM.

In this study we first identified peptides and neuropeptides from the brains of adult female *N. vespilloides* collected during three different behavioural and social states: virgin and solitary, actively caring and post-caring and solitary. We next examined abundances of neuropeptides in the different states and found that 8 of 18 changed in abundance in at least one state, with 7 increasing in abundance during parenting. Consistent with our hypothesis, these 7 are known to function in pathways of the behaviours that ethologists predicted change during the evolution of parenting: feeding, mating, aggression and social tolerance. Importantly, no neuropeptides that have functions outside of these behaviours changed in abundance. Our work supports the notion that ethological principles can be used to a priori identify candidate genetic pathways and molecules that influence complex behaviours.

## Results

**Identification of neuropeptides in *Nicrophorus vespilloides*.** We identified 133 peptides in the brains of *N. vespilloides* belonging to neuropeptide proteins (Supplementary Table 1). We found very few peptides identified in one state but not others. Specifically, actively parenting individuals exclusively displayed two peptides from FMRFamide (FMRFa; DKGHFLRF and GDLPANYE-MEEGYDRPT) and a single peptide from Neuropeptide-like 1 (NPLP-1; KESYDDDYYRMAAF). No *Apis*-NVP-like (NVP) peptides of the sequence FLNGPTRNNYYTLSELLGAAQQEQ NVPLYQRYVL were found in actively parenting samples.

These 133 peptides allowed us to identify 18 neuropeptide proteins that were present in at least one behavioural state (virgins, actively parenting and post-parenting individuals; Supplementary Table 1). Twelve were represented in all three behavioural states, while pheromone biosynthesis activating neuropeptide (PBAN), short neuropeptide F (sNPF) and natalisin (NTL) were absent in post-parenting individuals, diuretic hormone 47 ($DH_{47}$) was only found in actively parenting individuals, and crustean cardioactive peptide (CCAP) was only found in post-parenting individuals. Virgins showed a higher level of variability than the other two behavioural states. Ion transport peptide (ITP) was detected in a single biological replicate (a virgin), and is therefore not included in any further analyses.

**Changes in neuropeptides associated with parenting.** Having defined these neuropeptides, we tested for changes in the overall abundances of all neuropeptides across behavioural states using a multivariate analysis of variance (ANOVA). We found statistically significant differences in the overall abundance between the states ($F_{2,9} = 28.476$; $P = 0.0001$; Fig. 2, Table 1). The nature of this multivariate difference is best illustrated by creating linear combinations of the neuropeptides with principle components analysis. Five principle components had eigenvalues greater than 1 and are presented in Table 1. The first PC describes overall abundance and explains 41% of the variance, although SIFamide (SIFa) contributed very little to this vector and CCAP, consistent with being present in only one state, was opposite in sign. The second principle component explains 20% of the variance and, using a criterion of loadings of 0.3 or greater[22], contrasts changes in SIFa with myoinhibiting peptide (MIP) and RYamide (RYa). No obvious interpretation is suggested to us by this pattern. The remaining PCs describe 10% or less of the variance, and present no obvious interpretation of the contrasting loading for each neuropeptide. Consistent with the majority of the differences arising due to overall abundance, Fig. 2 illustrates that the three behavioural states separate primarily along the PC1 axis.

**Table 1 | Principal component analysis (PCA) of neuropeptide abundance of virgins, actively caring and post-caring *Nicrophorus vespilloides* females.**

| | Principal components | | | | |
| | PC1 | PC2 | PC3 | PC4 | PC5 |
|---|---|---|---|---|---|
| *Neuropeptides* | | | | | |
| NPLP-1 | 0.307 | − 0.203 | 0.034 | − 0.156 | − 0.184 |
| TK | 0.315 | − 0.061 | 0.231 | 0.110 | − 0.297 |
| NVP | 0.276 | 0.173 | 0.215 | 0.289 | 0.027 |
| $DH_{31}$ | 0.171 | 0.103 | 0.244 | − 0.578 | 0.131 |
| FMRFa | 0.274 | 0.182 | − 0.302 | − 0.260 | − 0.058 |
| ITG | 0.233 | − 0.291 | − 0.108 | − 0.340 | 0.185 |
| SIFa | 0.089 | − 0.475 | 0.174 | 0.068 | 0.149 |
| IDL | 0.221 | − 0.289 | 0.265 | 0.046 | 0.047 |
| SK | 0.281 | − 0.164 | − 0.385 | 0.144 | − 0.068 |
| MYO | 0.249 | − 0.226 | − 0.388 | − 0.031 | − 0.253 |
| MIP | 0.190 | 0.317 | − 0.079 | 0.267 | 0.296 |
| RYa | 0.189 | 0.363 | − 0.344 | − 0.152 | 0.068 |
| PBAN | 0.290 | 0.235 | 0.095 | − 0.046 | − 0.182 |
| NTL | 0.234 | 0.047 | − 0.112 | 0.356 | 0.406 |
| sNPF | 0.205 | 0.180 | 0.327 | − 0.221 | 0.381 |
| $DH_{47}$ | 0.201 | 0.239 | 0.291 | 0.147 | − 0.496 |
| CCAP | − 0.285 | 0.175 | − 0.027 | − 0.204 | − 0.218 |
| | | | | | |
| Eigenvalues | 7.013 | 3.387 | 1.859 | 1.588 | 1.246 |
| % Variance explained | 41.25 | 19.93 | 10.94 | 9.34 | 7.33 |

PCs with eigenvalues exceeding 1 are reported.

**Figure 2 | Principal component analysis of neuropeptide relative abundances.** Graph of the association between abundances and three non-parenting and parenting behavioural states of *Nicrophorus vespilloides* (red: virgin, black: actively parenting, blue: post-parenting). Principal component analysis (PCA) based on four biological replicates of each behavioural state, with eight individual brains pooled to form a biological sample. Ellipses show the 95% confidence area of each group.

We followed these multivariate analyses of variance (ANOVAs) with a priori defined univariate comparisons to examine how the relative abundances of specific neuropeptides changed. The results of individual ANOVAs are presented in Supplementary Table 1 and here we describe the Tukey–

Kramer honest significant difference *post hoc* pairwise of the behavioural states where the overall ANOVA was significant. In general, when neuropeptide abundance changed, it increased within actively parenting individuals. NPLP-1 had higher abundance in actively parenting compared with post-parenting ($P = 0.0063$), as did tachykinin (TK) ($P = 0.020$), FMRFa ($P = 0.0023$), sulfakinin (SK) ($P = 0.0087$), PBAN ($P = 0.023$), NVP ($P = 0.043$) and NTL ($P = 0.044$). FMRFa was also more abundant when individuals were actively parenting compared with virgins ($P = 0.011$). Sulfakinin (SK) had higher abundance in virgins ($P = 0.026$) compared with post-parenting, as did NTL ($P = 0.018$). CCAP differed from all others in having higher abundance in post-parenting compared with either virgins ($P = 0.046$) or actively parenting ($P = 0.046$).

Although not reaching the level of conventional statistical significance, two neuropeptides that showed a strong trend toward differential expression were RYa ($F_{2,9} = 4.033$, $P = 0.056$) and myosuppressin (MYO; $F_{2,9} = 3.611$, $P = 0.071$). Both were most highly expressed in actively parenting individuals. The remaining neuropeptides showed no strong trends (Supplementary Table 1).

## Discussion

Our goal was to test the prediction that the mechanisms involved in the evolution of parental care reside in pathways reflecting behavioural precursors predicted by ethological principles. Ethologists predict that parenting involves modification of pre-existing behavioural traits including mating, feeding, aggression, resource defence and social tolerance[2,3]. We suggest that this implies that the mechanistic underpinnings of these behaviours must also be altered. Specifically, we predicted that the timing of expression of neuropeptides associated with the behaviours will be altered, and that this will be reflected in the abundance of neuropeptides that influence mating, feeding, aggression and social tolerance in different behavioural states. To test this, we examined peptide abundance, with the prediction that the neuropeptides that have changed abundance during parenting function in feeding, mating, aggression and social interactions in organisms that do not display parental care. We profiled these changes from brains of the burying beetle *N. vespilloides*, which provides direct care by regurgitating food to dependent offspring. We identified 18 neuropeptides in the brain of *N. vespilloides*, which is consistent with other studies of non-model organisms[23–25]. Of these, the abundance of eight neuropeptides changed during parenting, all but one increasing during parenting. Although this is not evidence of causality, it is a strong correlation consistent with a priori predictions. Although conclusive evidence for our hypothesis requires functional manipulations or comparative analyses, these results support our initial prediction derived from how behaviour evolves.

Parenting across species typically involves a pause of mating, feeding others, appropriately directed aggression for defence and social interactions[2–4]. If our predictions are correct, then the neuropeptides that are more abundant when parenting will function in these behaviours in other taxa. Moreover, the neuropeptides that do not show changes in abundance should not have known functions in these behaviours. The eight neuropeptides that differed in abundance during parenting (Supplementary Table 1) support our prediction. In other taxa, both FMRFa[26], NTL[27] and SK[28] influence mating. Feeding behaviour and food intake are influenced by NVP[24] and SK[28,29]. Aggression and resource defence are influenced by TK[30,31] and SK[28]. NPLP-1 (ref. 32), TK[23,32] and PBAN[33] all influence social interactions. Of the 11 neuropeptides that were not differentially expressed, many have poorly understood functions (for example,

ITG, RYa, MIP, MYO[34–36]) or function outside the predicted pathways (CCAP, DH$_{31}$, DH$_{47}$, IDL, ITP[36]). Three of these neuropeptides have the potential to function in the predicted pathways were sNPF[34,36], and ITG-like[24], which influences feeding, and SIFa[34,37], which influences mating. Critically, there were no results consistent with the null hypothesis that pathways are unpredictable; none of the differentially expressed neuropeptides we identified in this study function solely outside the predicted pathways. Thus, although we do not identify every known neuropeptide, those that we can identify fit our prediction. As a corollary to our predictions, our results support the idea that like candidate gene studies[12], hypotheses about the pathways that are co-opted are likely to be more robust than hypotheses about specific neuropeptides when examining analogous behaviour in novel species.

Our study suggests three areas for further consideration to understand the mechanisms underlying parental care. First, we suggest that knowing or predicting the behavioural modules that provide the substrate for behavioural evolution provides insights into proximate mechanisms by also providing predicted pathways. Here we associated changes in protein abundance, but gene expression changes are also potentially predictable using this logic[19,38]. This can be tested further in other behaviours where the selective pressures and targets are known and therefore the underlying behavioural traits that are predicted to change can be identified a priori. Second, we provide information about specific neuropeptides that appear to underpin parental care and these can be examined in other subsocial organisms. Comparative studies will help us move beyond correlations. Finally, by specifying the behavioural and genetic pathways expected to be co-opted when parenting evolves, we can then identify particularly influential molecules that deserve further examination in *N. vespilloides*. Functional studies are desperately needed for organisms outside the genetic model species, and our work suggests several candidates. Among those neuropeptides we have identified, both tachykinin and sulfakinin influence nearly all the pathways thought to be co-opted during the evolution of parenting[28–32] and deserve further investigation in comparative or functional contexts.

## Methods

**Experimental design.** We used female *N. vespilloides* derived from an outbred colony we maintain at the University of Georgia, Athens. The colony was founded with beetles originally captured from Cornwall, UK and is subsidized yearly with new beetles from the same location. Thus, the colony is outbred. Larvae that disperse from a carcass were allocated to individual 9 mm diameter 4 cm deep circular containers filled with 2.5 cm potting soil. After emergence to adult, beetles were fed once weekly with decapitated mealworms *ad libitum*. Once larvae dispersed, larvae, pupae and adults had no further social interactions with other burying beetles until adults were paired for mating. We maintained all beetles in a common room at $22 \pm 1\,°C$ on a 15:9 h light:dark cycle.

To examine how neuropeptide expression changed with transitions of behavioural state, we collected age-matched females in three behavioural and social states: virgin (no social experience, no mating, no reproductive resource and no parenting), actively parenting (social experience, mated, reproductive resource and actively parenting), post-parenting (social experience, mated, reproductive resource and past parenting experience). Full descriptions of each behavioural state can be found in Roy-Zokan *et al.*[39] We collected virgins directly from their individual housing boxes. We collected actively parenting females directly from the carcass cavity where offspring are fed. We collected post-parenting females 9 days from the start of a breeding cycle after they had been isolated for 24 h. We collected all beetles at 19–22 days post-adult eclosion and all beetles were fed 1 day before their collection or before their pairing to standardize feeding status.

We performed dissections in ice-cold $1 \times$ phosphate-buffered saline (National Diagnostics, Atlanta, GA, USA) and completed them within 4 min. We placed single brains into 0.6 ml Eppendorf tubes with 30 μl of ice-cold acidified acetone extraction buffer (40:6:1 (v/v/v) acetone: $H_2O$: Concentrated HCl). We did not collect the retro-cerebral complex (corpora allata–corpora cardiaca). Once collected, we stored samples at $-80\,°C$ until extraction.

We pooled eight brains into a single biological replicate by removing brains and their associated acetone extraction buffer to a single 2.0 ml low protein binding

Vivacon 500 tubes (Sartorius AG, Göttingen, Germany). We pooled eight brains; this number was based on preliminary studies that confirmed that eight reliably provided sufficient material. We collected four biological replicates per behavioural state. We sonicated each biological replicate with a Sonicator S-4000 (Misonix, Farmingdale, NY, USA) fitted with a 1/8″ tip (#419) set to an amplitude of 20 for a total of 60 s sonication with 15 s pulses followed by 15 s rest on ice. We then centrifuged replicates at 16,000 g for 20 min at 4 °C with a 5810-R Eppendorf centrifuge. We collected the supernatant into a new Vivacon tube and repeated the extraction with the same volume of buffer and sonication protocol. We pooled and extracted all replicates at the same time without ordering. We stored samples at 4 °C until LC-MS/MS analysis.

We analysed our biological replicates with a Finnigan LTQ linear ion trap mass spectrometer (Thermo-Fisher Scientific, Waltham, MA, USA) and an 1100 Series Capillary LC system (Agilent Technologies, Santa Clara, CA, USA) with an ESI source with spray tips built in-house. The extraction buffer was vacuum-dried off of all biological replicates with a VirTis Benchtop K Lyophilizer (SP Scientific, Warminster, PA, USA) and biological replicates were suspended in 11 μl of buffer A (5% acetonitrile/0.1% formic acid/10 mM ammonium formate) and 8 μl of each replicate were injected into the LC column. The peptides were separated using a 200 μm × 150 mm HALO Peptide ES-C18 column packed with 5 μm diameter superficially porous particles (Advanced Materials Technology, Wilmington, DE, USA). The gradient used for each replicate was 5–75% buffer B (80% acetonitrile/ 0.1% formic acid/10 mM ammonium formate) for 120 min at a 2 μl min$^{-1}$ flow rate. The settings for the mass spectrometer included taking the five most intense ions from each full mass spectrum for fragmentation using collision-induced dissociation and the resulting MS/MS spectra were recorded. Our biological replicates from the three treatments were interspersed with each other for LC-MS/MS analysis. All chemicals were LC-MS or molecular biology grade.

**Neuropeptide identification and analysis.** We converted the resulting RAW spectra using Trans Proteomic Pipeline (Seattle Proteome Center, Seattle, WA, USA). MS/MS spectra were then imported into MASCOT (v2.2.2; MatrixScience, Boston, MA, USA) and searched against all annotated proteins from the published *N. vespilloides* genome[21] to produce peptide-spectrum-matching scores only. We set search parameters as: enzyme, none; fixed modifications, none; variable modifications as oxidation (M), acetyl (N terminus), pyroglutamic acid (N terminus glutamine), and amidation (C terminus); maximum post-translational modifications, 6; peptide mass tolerance, ± 1,000 p.p.m.; fragment mass tolerance, ± 0.6 Da, tolerances set by the machine.

We identified proteins with ProteoIQ (v2.6.03; default setting; Premier Biosoft, Palo Alto, CA, USA), which filters and uses MASCOT peptide-spectrum matching scores to statistically validate proteins identifications using the PeptideProphet Protein Probability scoring algorithm[40]. We identified proteins, peptides and assigned spectral counts using all biological replicates within each behavioural state. We only tallied the 'top-hit' for each spectrum as a further restriction on quantification. We also used ProteoIQ to estimate abundance of neuropeptides after the secondary validation of protein identities. This analysis produces a list of peptides assigned to each identified protein and from this we looked for qualitative differences in the presence/absence of peptides across the behavioural states for peptides that had at least three spectra and were not truncated forms of a larger observed peptide from a particular protein. We excluded peptides from proteins that were only observed in a single behavioural state. We then calculated normalized spectral abundance factor (NASFs) for all proteins within each biological replicate using the protein length for the NASF length correction factor[41]. Only peptides with at least two spectra within one biological replicate were quantified. Neuropeptide proteins were extracted from the overall protein list after establishing their identity within the published *N. vespilloides* gene set with a *Tribolium castaneum* neuropeptidome[42] augmented with neuropeptides identified and described from other insects[27,43–46]. We confirmed each neuropeptide's identity using NCBI's non-redundant insect protein database. We also assessed whether each neuropeptide had a predicted signal peptide using SignalP (v4.1; ref. 47) with a D-cutoff value of 0.34 (ref. 48).

To test the hypothesis that changes in neuropeptide expression can be predicted a priori, we first performed a multivariate ANOVA to establish that there was an overall difference in the neuropeptide composition between treatments. A significant multivariate analysis allows for univariate a priori contrasts using ANOVA without inflating the Type I error[22]. We followed this multivariate test with univariate tests (ANOVAs) for difference of individual neuropeptide abundance, testing for the effect of behavioural state on abundance. Where the ANOVA was significant, we performed *post hoc* tests of differences in the pairwise means of the behavioural states using Tukey–Kramer honest significant difference tests, which allow us to compare pairs of behavioural states while controlling for FDR. All statistical analyses were conducted with JMP Pro (v11.0.0, Cary, NC, USA). Raw abundances, the NASF values, of each neuropeptide from every sample were used to calculate composite abundances using principal component analysis. We used R (v3.3.1) using the prcomp function after scaling the raw abundance data for each detected peptide to mean zero and unit variance to calculate PCAs. Visualizations were prepared in R using ggbiplot (v0.90; github.com/vqv/ggbiplot).

**Data availability.** Raw mass spectral data are available at ProteomeXchange (PXD005460; proteomecentral.proteomexchange.org).

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

## Acknowledgements

We thank Elizabeth McKinney for technical support. This research was supported by funding to A.J.M. from the US National Science Foundation (IOS-1354358) and the University of Georgia's Office of the Vice-President for Research.

## Author contributions

C.B.C., R.B.M., R.O. and A.J.M. designed the experiment. C.B.C. gathered and extracted the biological samples with assistance from R.B.M. M.J.B. and R.O. conducted the LC-MS/MS analysis. C.B.C. and A.J.M. analysed the results with guidance from M.J.B. and R.O. on peptide and neuropeptide identification. C.B.C. and A.J.M. wrote the manuscript with input from all the authors. All the authors approved the final manuscript.

## Additional information

**Competing financial interests:** The authors declare no competing financial interests.

