## [Peer Review File · Nature Communications]

Reviewers' comments:

Reviewer #1 (Remarks to the Author):

This is a very nice experiment by Cunningham et al, where the authors hypothesize that neuropeptides that influence feeding, mating, aggression, and increased tolerance for social internations will differ in abundance in parenting individuals. I have a major conceptual concern and then a few specific comments that I hope will aid the authors in improving this manuscript.

General comments:

The authors present a very interesting hypothesis - that in the evolution of parental care, a suite of behaviors (aggression, sociality, resource defense) must be altered. The authors then set out to test if neuropeptides change in abundance as female beetles transition from solitary virgins to being a parent and then solitary post-parenting. My major concern is that the experiment presented is not sufficient to accept or reject this very broad hypothesis. The hypothesis is rooted in an evolutionary question and examining the ontogeny of parental care in a single sex in a single species without functional manipulations or other comparisons does not strongly test the overarching hypothesis. The authors conclude that changes in neuropeptide abundance are implicated in the evolution of parental care, but are changes in neuropeptide abundance within a single sex surprising? To make conclusions about the role of the neuropeptides in the evolution of parental care, I believe the authors either need to 1) perform functional manipulations (blocking neuropeptide action) within parenting females or 2) take advantage of a "natural manipulation" by comparing neuropeptide abundance in a non-parental beetle.

Specific comments:

ABSTRACT

-Expression vs Abundance. This is just semantics, but I think of measuring RNA expression and protein abundance. Please be consistent (expression vs abundance) throughout the manuscript.

- Line 35. Missing a period at the end of this sentence.

INTRODUCTION

-lines 58-62. Please break this sentence up into two or make transition into list easier.

-Line 80. I believe this references is misnumbered, as ref 14 is a paper about galanin.

-Line 81. I believe this reference is misnumbered. It looks like many of the remaining references are one citation off in numbering and this will be my last comment about that.

RESULTS

- Are there protein names that go along with these peptides so as to not have a list of undefined acronyms? This may give them a little more meaning to the reader.

- Line 116. Protein not proteins

DISCUSSION

- Lines 173-175. Are neuropeptides important for parenting only if they differed compared to BOTH the virgin and post-parenting comparison? If they only differed from one, what are the implications for what these neuropeptides are doing in terms of behavior? It is not clear what the results of the Tukey's HSD test are.
- Line 184. Incomplete parentheses.
- Line 190. The word homologous suggests a common origin. Perhaps "analogous behaviour" would be more appropriate depending on the comparison the authors are suggesting, which is not clear.

METHODS

- Line 233. replicate (singular)
- Where the post hoc tests corrected for multiple testing and how?
- There is no description of the PCA analysis. What data was included in this analysis?

Table 2.

- What is a non-functional peptide and how do you know this?
- I assume the stats in the corners are from the ANOVAs. Where are the results from the posthoc tests? When something says up post parenting, does mean compared to both of the other groups or just one?

Reviewer #2 (Remarks to the Author):

The authors explore the mechanisms that could have led to the evolution of parental care and predict that parenting will involve changes in gene expression influencing feeding, mating, aggression, and tolerance for social interactions. Using multivariate analyses, they find differences in relative abundance of all neuropeptides across different behavioral states (virgin, active parents, post-parenting). They then identify some neuropeptides that tend to have greater expression when individuals are actively parenting.

This is a relatively straightforward and elegant study that addresses a timely and broad question from a creative mechanistic perspective. The results are intriguing and consistent with a priori predictions. While the results are not largely unexpected, the focus on the mechanisms that likely allowed for the evolution of care is highly novel, and I believe that this study will motivate future research on the topic.

Reviewer #3 (Remarks to the Author):

The authors present a proteomics study aimed at dissecting the mechanistic basis of parental care in burying beetles. Authors find that certain neuropeptides involved in sub-categories of behaviors associated with parental care, e.g., aggression, mating, etc. are differentially regulated comparing solitary, parenting, and post-parenting solitary beetles.

There are certain parts of this study that are conceptually interesting – e.g., dissecting parental care into sub-behaviors that can be used to develop a priori hypotheses about the evolution of parental care itself. The proteomics approach is not novel but is well justified here, and presents an important alternative to more common transcriptomic approaches. However, I think more needs to be done to justify the impact of the study, and to tie together the approach and the results. I have made many comments to this effect. The quality of the data presentation could be improved, and some more detailed analyses that follow from the predictions laid out in the introduction would make the study more compelling (see below).

L67 – I found the wording of this paragraph a little confusing – it, to a degree, confounds sequence level changes and changes in gene expression patterns, both of which are likely involved in the evolution of parental care (but addressing each requires different approaches, and it is not clear yet which you are going to take in the current study). Maybe you could tie this to your specific approach to clarify. You also plant a seed about pleiotropy, and it is not clear exactly how that relates to your approach either.

L87 – Here again I was a little confused. I think this paragraph could be reworded. As written it sort of confounds approach (transcriptomics, proteomics) with target (neuropeptides). Maybe just clarifying that you expect a role for neuropeptides via a non-transcriptional mechanism? I think you also need to connect this to the gene expression/regulating comments above – transcriptomics may not capture neuropeptide activity, but what does that mean about their evolution? Some of this may be better in the discussion.

I do think the point about neuropeptide assessment via transcriptomics is highly valuable, as many similar studies fail to find changes in gene expression related to canonical neurotransmitters/peptides.

L96 – should be “a” solitary?

I think you can do more in the introduction to make the case for the modular nature of behaviors associated with parental care, and perhaps provide a clearer statement of the insights to be gained from evaluating the transcriptomic/proteomic basis of these sub-behaviors, which likely involve pleiotropic gene regulation. There is a lot of interesting information here but it does not tie together in a clear way to point towards the major question.

L116 – “each neuropeptide proteins” typo

L118 – is FMRFa defined somewhere? Same for L120. Since methods are at the end provide more context for the results.

L125 – please redefine the states here, and perhaps hypotheses/predictions again

L133-135 – these two sentences sound like they are saying the same thing. Please clarify

L136 – no apostrophe in ANOVAs

L140-143 – here you have two p-values listed for each comparison. I am assuming one is from the ANOVA and one is from a post-hoc test? Can you list the test with the p-value?

L131-153 – paragraph is highly repetitive. Maybe you can cluster the results on some basis, e.g., by behavior (which of these peptides is associated with aggression etc) or by result (which of these peptides is differentially expressed from solitary to parenting versus solitary to post-parenting, etc? Also, you do not need to state that each is differentially expressed overall and across comparisons. You can just state the latter and list p-values for the full ANOVA.

Figure 2 should be discussed more explicitly in the results portion (which gets at my confusion above, L133-135). How much variance explained by each component, can you tell what each means, etc. Even simply stating that the PCA clearly separated treatments is nice to know.

L167 – your approach suggests you have predictions about which neuropeptides are differentially regulated across these different behavioral states (i.e., because of the role of aggression and other sub-behaviors in parental care), but you do not present or discuss the results in this framework. Instead, you talk more generally about differences across states. It would be more powerful to have an a priori prediction about specific sets of neuropeptides associated with various behavioral modules, and then test the involvement of each of these sets with an enrichment analysis. I realize you may not have the power to do that, but the paper would be a more interesting contribution if your analysis followed with this modular hypothesis. The hypothesis does not really match the data analysis at the moment.

Do burying beetles have an oxytocin homolog? It would be interesting to look explicitly at “bonding” neuropeptides (i.e., those ID’d in the mammalian literature), even if the result is that there are few that show activity.

L177-190 – the information in this paragraph is interesting, but the writing is a bit unclear. I think you should give more context for your results. The reader will not necessarily know what all of these acronyms mean and why they are important. Also, I am not sure how to interpret your statement about non-significant results. How many pathways are you considering? Are they interrelated? Maybe a figure, showing a pathway map and direction of expression change, would help.

L194 – typo “is” versus “in”

L193 – I am not sure what you mean by “selective pressures”. It seems it would be more accurate to say that parsing out behavioral modules sub-serving parental care can provide predictions about associated genes. I’m not sure it is useful to invoke selection here.

L198 – the need for functional studies is separate from providing new predictions about the genes involved in parental care...

L202 – I think you could discuss some of these points more thoroughly. There is a large and growing literature on tachykinin, and no papers are cited here.

The discussion is thin. If the journal allows, it may be more productive to merge your results and discussion, as this will have the added benefit of providing more context for your results.

L233 – typo

Was your LC-MS/MS experiment designed to capture all possible peptides or to target certain peptides? Usually these types of analyses are not quite as “all inclusive” as transcriptomics studies (to which you compare this study in the introduction and discussion).

Do you have a justification for using 8 brains/biological replicate?

Reviewer #4 (Remarks to the Author):

In general, this is a well performed study to determine how complex suites of peptides change during different life stages and social behaviors.

One of the underlying assumptions is that the specific peptides have some “original” role and the behavioral roles described here are a new function in parenting. It is hard to understand this in terms of which role is “original”. When one thinks of the classical transmitters, one can state that 5-HT is involved in feeding, but could list systems throughout the animal that have co-opted 5-HT signaling, whatever that means. This is also true for peptides. Thus, I am not sure this is co-opting a peptide, especially as few if any peptides have a single behavior “function”. In this case, they see changes in a complex suite of peptides as they are looking at complex social behaviors.

Mass spectrometry. There is an active field of neuropeptidomics (for insects, mammals, and other models), and this field has developed a range of MS approaches and protocols to yield high quality data. The current work appears somewhat independent of the field of neuropeptide characterization (which was not well cited). Issues include the relatively poor requirements for peptide identification (mass tolerance, etc.). There are few details on the identification requirements. Do all the identified peptide precursors have signal sequences? Peptide amidation requires the next amino acid is a glycine (G). (Said differently, if the next amino acid is not a glycine, the peptide cannot be amidated). Search engines like MASCOT do not take this into account, and so all amidations need to be manually checked. There are several cases in table 1 where amidation is reported that appears to be in error given this requirement. These potential misidentifications exacerbate the identification issues listed above. What kind of FDR was used in identifying significantly changed peptides? It appears

that after the initial tests, pairwise comparisons are made without any corrections. In other words, more discussions of the dataset generated as part of this work would be useful.

Reviewers' comments:

Reviewer #1 (Remarks to the Author):

This is a very nice experiment by Cunningham et al, where the authors hypothesize that neuropeptides that influence feeding, mating, aggression, and increased tolerance for social internations will differ in abundance in parenting individuals. I have a major conceptual concern and then a few specific comments that I hope will aid the authors in improving this manuscript.

General comments:

The authors present a very interesting hypothesis - that in the evolution of parental care, a suite of behaviors (aggression, sociality, resource defense) must be altered. The authors then set out to test if neuropeptides change in abundance as female beetles transition from solitary virgins to being a parent and then solitary post-parenting. My major concern is that the experiment presented is not sufficient to accept or reject this very broad hypothesis. The hypothesis is rooted in an evolutionary question and examining the ontogeny of parental care in a single sex in a single species without functional manipulations or other comparisons does not strongly test the overarching hypothesis. The authors conclude that changes in neuropeptide abundance are implicated in the evolution of parental care, but are changes in neuropeptide abundance within a single sex surprising? To make conclusions about the role of the neuropeptides in the evolution of parental care, I believe the authors either need to 1) perform functional manipulations (blocking neuropeptide action) within parenting females or 2) take advantage of a "natural manipulation" by comparing neuropeptide abundance in a non-parental beetle.

We agree and have modified our presentation to acknowledge this point. We have added a statement addressing the need for experimental manipulations or comparative studies in lines 261-263, and again in the concluding paragraph of the discussion. We are careful to note that our study is suggestive but not conclusive without such additional information.

Specific comments:

ABSTRACT

-Expression vs Abundance. This is just semantics, but I think of measuring RNA expression and protein abundance. Please be consistent (expression vs abundance) throughout the manuscript.

Done. Unless discussing gene expression, we now use abundance throughout when describing our peptide data and analyses.

- Line 35. Missing a period at the end of this sentence.

Done

INTRODUCTION

-lines 58-62. Please break this sentence up into two or make transition into list easier.

Done (see lines

-Line 80. I believe this references is misnumbered, as ref 14 is a paper about galanin.

-Line 81. I believe this reference is misnumbered. It looks like many of the remaining references are one citation off in numbering and this will be my last comment about that.

All references have been checked and corrected.

RESULTS

- Are there protein names that go along with these peptides so as to not have a list of undefined acronyms? This may give them a little more meaning to the reader.

We now provide the peptide names alongside their acronyms where there first appear in the results. (See lines 142, 144, 151-155, etc.)

- Line 116. Protein not proteins

Corrected.

DISCUSSION

- Lines 173-175. Are neuropeptides important for parenting only if they differed compared to BOTH the virgin and post-parenting comparison? If they only differed from one, what are the implications for what these neuropeptides are doing in terms of behavior? It is not clear what the results of the Tukey's HSD test are.

We have clarified this and, also in response to reviewer 4, we have detailed better that our prediction was just that the peptides that changed would be those involved in feeding, aggression, mating or social interactions.

- Line 184. Incomplete parentheses.

Corrected

- Line 190. The word homologous suggests a common origin. Perhaps "analogous behaviour" would be more appropriate depending on the comparison the authors are suggesting, which is not clear.

We have changed this word to “analogous” (line 230). We are trying to say that (e.g.) predicting “feeding” pathways will be involved in parenting is more likely to be supported than predicting NPF versus sNPF.

METHODS

- Line 233. *replicate (singular)*

Corrected

- *Where the post hoc tests corrected for multiple testing and how?*

We have clarified this (lines 174-175; 334-342). As we now state in the ms, the MANOVA protects against inflated type I error that may accompany multiple ANOVA tests if the multivariate analysis is (overall) significant.

- *There is no description of the PCA analysis. What data was included in this analysis?*

Thank you for pointing this out – it was an oversight on our part. We have expanded this description (Line 343-347). We now describe the results of the PCA more fully (line 161-172).

Table 2.

- *What is a non-functional peptide and how do you know this?*

We now indicate (table 1) where there are possible modifications and C-terminus amidation, and where amidation is not well-supported (see also response to reviewer 4).

- *I assume the stats in the corners are from the ANOVAs. Where are the results from the posthoc tests? When something says up post parenting, does mean compared to both of the other groups or just one?*

We now clarify these are ANOVA results in Table 1. Post-hoc results are given in the text (Lines 179-185), and comparisons are to just one (i.e., pairwise comparisons of behavioral states) and this is indicated in the text (Lines 175-178).

Reviewer #2 (Remarks to the Author):

The authors explore the mechanisms that could have led to the evolution of parental care and predict that parenting will involve changes in gene expression influencing feeding, mating, aggression, and tolerance for social interactions. Using multivariate analyses, they find differences in relative abundance of all neuropeptides across different behavioral states (virgin, active parents, post-

parenting). They then identify some neuropeptides that tend to have greater expression when individuals are actively parenting.

This is a relatively straightforward and elegant study that addresses a timely and broad question from a creative mechanistic perspective. The results are intriguing and consistent with a priori predictions. While the results are not largely unexpected, the focus on the mechanisms that likely allowed for the evolution of care is highly novel, and I believe that this study will motivate future research on the topic.

Thank you – this was exactly the response we had hoped for.

Reviewer #3 (Remarks to the Author):

The authors present a proteomics study aimed at dissecting the mechanistic basis of parental care in burying beetles. Authors find that certain neuropeptides involved in sub-categories of behaviors associated with parental care, e.g., aggression, mating, etc. are differentially regulated comparing solitary, parenting, and post-parenting solitary beetles.

There are certain parts of this study that are conceptually interesting – e.g., dissecting parental care into sub-behaviors that can be used to develop a priori hypotheses about the evolution of parental care itself. The proteomics approach is not novel but is well justified here, and presents an important alternative to more common transcriptomic approaches. However, I think more needs to be done to justify the impact of the study, and to tie together the approach and the results. I have made many comments to this effect. The quality of the data presentation could be improved, and some more detailed analyses that follow from the predictions laid out in the introduction would make the study more compelling (see below).

Thank you; the suggested changes help and also address some of the concerns raised by reviewer 1 and 4.

L67 – I found the wording of this paragraph a little confusing – it, to a degree, confounds sequence level changes and changes in gene expression patterns, both of which are likely involved in the evolution of parental care (but addressing each requires different approaches, and it is not clear yet which you are going to take in the current study). Maybe you could tie this to your specific approach to clarify. You also plant a seed about pleiotropy, and it is not clear exactly how that relates to your approach either.

We have revised this paragraph (Lines 73-92) with the hope that it is now clearer. In general, we provide more specifics. As suggested by reviewer 1, we are really talking about protein abundance and gene expression and keeping these clear by using “abundance” in relation to proteins helps.

L87 – Here again I was a little confused. I think this paragraph could be reworded. As written it sort of confounds approach (transcriptomics, proteomics) with target (neuropeptides). Maybe just clarifying that you expect a role for neuropeptides via a non-transcriptional mechanism? I think you also need to connect this to the gene expression/regulating comments above – transcriptomics may not capture neuropeptide activity, but what does that mean about their evolution? Some of this may be better in the discussion.

I do think the point about neuropeptide assessment via transcriptomics is highly valuable, as many similar studies fail to find changes in gene expression related to canonical neurotransmitters/peptides.

As suggested, we have reworded this paragraph (Line 94-111). The main change was to start with the point about transcriptomics being a poor way to assess neuropeptides, which was in fact the point of the paragraph anyway. We have now elevated this to the topic sentence (Line 94-95) instead of burying it at the end of the paragraph. The two paragraphs together now move from transcriptomics to proteomics and indicate their utility and complementarity, providing rationale for this study.

L96 – should be “a” solitary?

Corrected

I think you can do more in the introduction to make the case for the modular nature of behaviors associated with parental care, and perhaps provide a clearer statement of the insights to be gained from evaluating the transcriptomic/proteomic basis of these sub-behaviors, which likely involve pleiotropic gene regulation. There is a lot of interesting information here but it does not tie together in a clear way to point towards the major question.

We agree. We have taken this suggestion to heart when revising the second and third paragraphs to better make the points suggested.

L116 – “each neuropeptide proteins” typo

Corrected

L118 – is FMRFa defined somewhere? Same for L120. Since methods are at the end provide more context for the results.

Thank you for the suggestion – these are now spelled out on first use throughout the results. See response to reviewer 1 (and Lines 142, 144, 151-155, etc.)

L125 – please redefine the states here, and perhaps hypotheses/predictions

again

Done

L133-135 – these two sentences sound like they are saying the same thing. Please clarify

They were redundant and we deleted the second sentence.

L136 – no apostrophe in ANOVAs

Corrected

L140-143 – here you have two p-values listed for each comparison. I am assuming one is from the ANOVA and one is from a post-hoc test? Can you list the test with the p-value?

Corrected – we now specify the comparison associated with each p-value (Lines 179-185).

L131-153 – paragraph is highly repetitive. Maybe you can cluster the results on some basis, e.g., by behavior (which of these peptides is associated with aggression etc) or by result (which of these peptides is differentially expressed from solitary to parenting versus solitary to post-parenting, etc? Also, you do not need to state that each is differentially expressed overall and across comparisons. You can just state the latter and list p-values for the full ANOVA.

We agree that the ANOVA results are numerous and therefore tedious to read. We have made two changes to improve readability. First, the ANOVA results (F-tests, P-values) are given in Table 1 and therefore it is redundant to present these in the text. We now just present the Tukey-Kramer HSD post-hoc pairwise analyses in the text. Second, we group the results by the comparison, highlighting that for most the parenting state has highest abundance. See Lines 174-185.

Figure 2 should be discussed more explicitly in the results portion (which gets at my confusion above, L133-135). How much variance explained by each component, can you tell what each means, etc. Even simply stating that the PCA clearly separated treatments is nice to know.

We agree and it was an oversight not to do so. We now do this immediately after the MANOVA, keeping the multivariate analyses together (Lines 161-172). This, we think, helps clarify the point you make that there are two analyses here – the multivariate investigation of overall differences (without examining specific neuropeptides) and then the univariate analysis of each neuropeptide.

L167 – your approach suggests you have predictions about which neuropeptides are differentially regulated across these different behavioral states (i.e., because of the role of aggression and other sub-behaviors in parental care), but you do not present or discuss the results in this framework. Instead, you talk more generally about differences across states. It would be more powerful to have an a priori prediction about specific sets of neuropeptides associated with various behavioral modules, and then test the involvement of each of these sets with an enrichment analysis. I realize you may not have the power to do that, but the paper would be a more interesting contribution if your analysis followed with this modular hypothesis. The hypothesis does not really match the data analysis at the moment.

We have clarified the a-priori predictions and better tie the writing here to that in the introduction. The first paragraph is re-written to highlight the predictions (Lines 61-71). The approach suggested is an interesting and valuable alternative experiment, but one that we did not and cannot yet perform.

Do burying beetles have an oxytocin homolog? It would be interesting to look explicitly at “bonding” neuropeptides (i.e., those ID’d in the mammalian literature), even if the result is that there are few that show activity.

They do and we agree. Unfortunately, inotocin (the oxytocin homolog) did not turn up in our proteomic analysis. We are therefore investigating this independently. Stay tuned.

L177-190 – the information in this paragraph is interesting, but the writing is a bit unclear. I think you should give more context for your results. The reader will not necessarily know what all of these acronyms mean and why they are important. Also, I am not sure how to interpret your statement about non-significant results. How many pathways are you considering? Are they interrelated? Maybe a figure, showing a pathway map and direction of expression change, would help.

We have provided more context, making it more explicit how we test our predictions with these results, and how they fit into the overall hypothesis (Lines 214-218; .

L194 – typo “is” versus “in”

Corrected (deleted “is general”)

L193 – I am not sure what you mean by “selective pressures”. It seems it would be more accurate to say that parsing out behavioral modules sub-serving parental care can provide predictions about associated genes. I’m not sure it is useful to invoke selection here.

We have reworded and clarified this (Lines 234-235).

L198 – the need for functional studies is separate from providing new predictions about the genes involved in parental care...

True, and we intended to imply that knowing the genes facilitate functional studies. We have clarified this (Lines 243-247). See also response to reviewer 4.

L202 – I think you could discuss some of these points more thoroughly. There is a large and growing literature on tachykinin, and no papers are cited here.

We have added the missing references (Line 246). We agree that there is more we could speculate upon, although our intent here was to stimulate others to think more carefully and fully about the issues. We therefore have refrained from expanding this concluding paragraph and yet attempting to clearly signpost future studies (lines 233-247).

The discussion is thin. If the journal allows, it may be more productive to merge your results and discussion, as this will have the added benefit of providing more context for your results.

The revised discussion is much better focused and detailed (and much better), in response to all of the reviewer comments.

L233 – typo

Corrected

Was your LC-MS/MS experiment designed to capture all possible peptides or to target certain peptides? Usually these types of analyses are not quite as “all inclusive” as transcriptomics studies (to which you compare this study in the introduction and discussion).

Correct, and we have expanded the methods to indicate how neuropeptides were targeted (lines 313-319; 328-332).

Do you have a justification for using 8 brains/biological replicate?

We now indicate in the paper that this was determined by preliminary studies, working to find the fewest number of brains that provided consistent yields across samples (Lines 278-279).

Reviewer #4 (Remarks to the Author):

In general, this is a well performed study to determine how complex suites of peptides change during different life stages and social behaviors.

We appreciate the positive comments.

One of the underlying assumptions is that the specific peptides have some “original” role and the behavioral roles described here are a new function in parenting. It is hard to understand this in terms of which role is “original”. When one thinks of the classical transmitters, one can state that 5-HT is involved in feeding, but could list systems throughout the animal that have co-opted 5-HT signaling, whatever that means. This is also true for peptides. Thus, I am not sure this is co-opting a peptide, especially as few if any peptides have a single behavior “function”. In this case, they see changes in a complex suite of peptides as they are looking at complex social behaviors.

We have worked to clarify our point here (especially lines 76-85; see also response to reviewer 3 regarding the revision to the introduction). Our hypothesis is that there are ancestral behaviors that are affected by a suite of neuropeptides. We suggest that these neuropeptides gain additional roles (such as “parenting”) when new behavior evolves because this behavior is co-opted, and this occurs by changing the timing or level of expression of the genes that produce neuropeptides. We now include a hypothetical example (Lines 85-88), which we hope makes the point clearer.

Mass spectrometry. There is an active field of neuropeptidomics (for insects, mammals, and other models), and this field has developed a range of MS approaches and protocols to yield high quality data. The current work appears somewhat independent of the field of neuropeptide characterization (which was not well cited). Issues include the relatively poor requirements for peptide identification (mass tolerance, etc.). There are few details on the identification requirements. Do all the identified peptide precursors have signal sequences? Peptide amidation requires the next amino acid is a glycine (G). (Said differently, if the next amino acid is not a glycine, the peptide cannot be amidated). Search engines like MASCOT do not take this into account, and so all amidations need to be manually checked. There are several cases in table 1 where amidation is reported that appears to be in error given this requirement. These potential misidentifications exacerbate the identification issues listed above. What kind of FDR was used in identifying significantly changed peptides? It appears that after the initial tests, pairwise comparisons are made without any corrections. In other words, more discussions of the dataset generated as part of this work would be useful.

We appreciate these comments and have expanded our methods to provide greater detail. Many of these comments were reflected in the reviewer 3 suggested revisions as well.

We have increased our methods to clarify how our protocol validated protein identifications (Lines 307-308; 313-319). We have added to the methods section a discussion of how we identified the neuropeptides in the published *N*.

vespilloides gene set (Lines 306-307; 328-332). We have manually checked amidation events. All of the peptides followed by glycine do have spectra that support amidation, while 5/9 of the peptides not followed by glycine did not have strong support for being amidated. We have added an asterisk beside each of the questionably amidated peptide in table 1 to indicate the weak support for amidation. Finally, we clarify the fact that ANOVA a-priori comparisons FDR are controlled by first performing MANOVA (Lines 174-175; 334-342).

REVIEWERS' COMMENTS:

Reviewer #1 (Remarks to the Author):

The authors did a good job revising the manuscript, as it is now much more clear. It is very conceptually interesting, nicely explained, with correlative data to strengthen their a priori predictions about the evolution of a complex social behavior.

Reviewer #3 (Remarks to the Author):

In general, the manuscript is much improved. Here I provide a few additional comments that I think can be easily addressed in a revision. I think the most substantial comments I made here should be addressed as caveats somewhere in the manuscript, probably the discussion (see below).

L105 – typo – casual/causal

L140 – this sounds like you mean you found no differences in protein isoforms across states (I do not think that is what you mean...)

L142-146 – does this finding address a specific a priori hypothesis? Seems like it does as worded, but maybe you are just generally noting that these peptides were associated with your behavioral states.

L148 – these peptides? Which peptides?

L199 – I would focus on just the neuropeptide predictions since that is what you have evaluated here.

L202-204 – Moreover, should the nature of the timing of these expressed behaviors/peptides differ? For instance, parental care requires all traits to occur simultaneously – is the expectation that in the solitary condition, they do NOT overlap?

L215 – typo – “function (s)”

L215 – One thing I am struggling with is developing a null or alternative hypothesis to the one you have presented. Is the alternative that parenting involves a completely novel cluster of neuropeptides? These are the typical alternatives presented (conserved versus novel genes – though these are not mutually exclusive hypotheses). However, are these alternatives realistic, or can they be evaluated rigorously at the neuropeptide level given the overlapping functions of many neuropeptides, and the limited number of behaviorally relevant neuropeptides (compared, e.g., to the number of genes functionally expressed in association with a behavioral phenotype, which is often in the 100's)? Another way to look at an alternative could be related to how exactly these peptides/traits are expressed in the solitary condition (see timing comment above). That there should be some overlap makes sense, and you find some evidence for that here, but it is hard to rigorously test (especially without enough elements to do some sort of enrichment statistical comparison, and without a clear alternative). I think these caveats should be addressed at least in the discussion.

L225 – “Critically...” but again, how rigorous is this expectation really, given the high number of possible behavioral modules you are considering here (and the number of pathways underlying them)?

L234-236 – I found this sentence very confusing.

Reviewer #4 (Remarks to the Author):

The authors were responsive to the numerous reviewer suggestions. Their changes have resolved the issues and strengthened their presentation.

As a minor comment, their response to the amidation comments appears strange. If the next amino is not a glycine, the peptide is not amidated. Thus, the modification as listed is wrong. Stating that in some of these cases, the amidation is not well supported does not appear the appropriate answer when the biology tells us the assignment cannot be correct.

Response to REVIEWERS' COMMENTS:

Reviewer #1 (Remarks to the Author):

The authors did a good job revising the manuscript, as it is now much more clear. It is very conceptually interesting, nicely explained, with correlative data to strengthen their a priori predictions about the evolution of a complex social behavior.

Thank you.

Reviewer #3 (Remarks to the Author):

In general, the manuscript is much improved. Here I provide a few additional comments that I think can be easily addressed in a revision. I think the most substantial comments I made here should be addressed as caveats somewhere in the manuscript, probably the discussion (see below).

L105 – typo – casual/causal

Corrected

L140 – this sounds like you mean you found no differences in protein isoforms across states (I do not think that is what you mean...)

Edited for clarity (changed to: “We found very few peptides identified in one state but not others.”)

L142-146 – does this finding address a specific a priori hypothesis? Seems like it does as worded, but maybe you are just generally noting that these peptides were associated with your behavioral states.

This is now clarified by the above edit, and by connecting the two statements with the addition of “Specifically,” to this sentence.

L148 – these peptides? Which peptides?

Now specified (all 133).

L199 – I would focus on just the neuropeptide predictions since that is what you have evaluated here.

Changed “gene” to “neuropeptide” and clarified.

L202-204 – Moreover, should the nature of the timing of these expressed behaviors/peptides differ? For instance, parental care requires all traits to occur simultaneously – is the expectation that in the solitary condition, they do NOT overlap?

This is not a prediction and so we do not include it. We don't really have predictions for the solitary state other than it will differ from a parenting state.

L215 – typo – “function (s)”

Corrected

L215 – One thing I am struggling with is developing a null or alternative hypothesis to the one you have presented. Is the alternative that parenting involves a completely novel cluster of neuropeptides? These are the typical alternatives presented (conserved versus novel genes – though these are not mutually exclusive hypotheses). However, are these alternatives realistic, or can they be evaluated rigorously at the neuropeptide level given the overlapping functions of many neuropeptides, and the limited number of behaviorally relevant neuropeptides (compared, e.g., to the number of genes functionally expressed in association with a behavioral phenotype, which is often in the 100's)? Another way to look at an alternative could be related to how exactly these peptides/traits are expressed in the solitary condition (see timing comment above). That there should be some overlap makes sense, and you find some evidence for that here, but it is hard to rigorously test (especially without enough elements to do some sort of enrichment statistical comparison, and without a clear alternative). I think these caveats should be addressed at least in the discussion.

The question of an alternative is interesting, but we do not discuss it here as the alternatives we can imagine are not mutually exclusive to what we find and suggest here. I think it is probably that novel genes are involved but, as suggested, this isn't really mutually exclusive and so it isn't clear that it is an alternative. We now make the null explicit – that genes NOT involved in expected behavioral precursors will show altered abundance – this is in line 347-348.

L225 – “Critically...” but again, how rigorous is this expectation really, given the high number of possible behavioral modules you are considering here (and the number of pathways underlying them)?

As pointed out above, this is the null, which we now make explicit. The fact that there was no contradictory result we feel is critical.

L234-236 – I found this sentence very confusing.

We have clarified by adding another sentence to expand the idea.

Reviewer #4 (Remarks to the Author):

The authors were responsive to the numerous reviewer suggestions. Their changes have resolved the issues and strengthened their presentation.

Thank you.

As a minor comment, their response to the amidation comments appears strange. If the next amino is not a glycine, the peptide is not amidated. Thus, the modification as listed is wrong. Stating that in some of these cases, the amidation is not well supported does not appear the appropriate answer when the biology tells us the assignment cannot be correct.

We have now removed this statement from table 1, which was indicated with an asterisk and footnote, so that readers can evaluate for themselves our categorization of amidation. We (the biochemists in the group) respectfully agree to disagree with the referee regarding our classification of amidation, but this is minor and we are happy to simply remove the potentially confusing footnote. We appreciate that the reviewer sees this as minor, as do we, for it is not a central part of the study or paper. Finally, we will deposit all of the raw mass spectral data in ProteomeXchange.org, which should certainly allow anyone to check our classification.